# Biocompatible Materials Based on Plasticized Poly(lactic acid), Chitosan and Rosemary Ethanolic Extract I. Effect of Chitosan on the Properties of Plasticized Poly(lactic acid) Materials

**DOI:** 10.3390/polym11060941

**Published:** 2019-05-30

**Authors:** Cornelia Vasile, Elena Stoleru, Raluca Nicoleta Darie-Niţa, Raluca Petronela Dumitriu, Daniela Pamfil, Liliana Tarţau

**Affiliations:** 1Department of Physical Chemistry of Polymers, “Petru Poni” Institute of Macromolecular Chemistry, Romanian Academy, 41A Gr. Ghica Voda Alley, 700487 Iasi, Romania; elena.paslaru@icmpp.ro (E.S.); darier@icmpp.ro (R.N.D.-N.); rdumi@icmpp.ro (R.P.D.); pamfil.daniela@icmpp.ro (D.P.); 2“Grigore T. Popa” University of Medicine and Pharmacy Iasi, 16 University Street, 700115 Iasi, Romania; lylytartau@yahoo.com

**Keywords:** poly(lactic acid), chitosan, rosemary extract, natural additives, biocomposites

## Abstract

The purpose of the present study is to develop new multifunctional environmentally friendly materials having applications both in medical and food packaging fields. New poly(lactic acid) (PLA)-based multifunctional materials containing additives derived from natural resources like chitosan (CS) and rosemary extract (R) were obtained by melt mixing. Each of the selected components has its own specific properties such as: PLA is a biodegradable thermoplastic aliphatic polyester derived from renewable biomass, heat-resistant, with mechanical properties close to those of polystyrene and polyethylene terephthalate, and CS offers good antimicrobial activity and biological functions, while R significantly improves antioxidative action necessary in all applications. A synergy of their combination, an optimum choice of their ratio, and processing parameters led to high performance antimicrobial/antioxidant/biocompatible/environmentally degradable materials. The polyethylene glycol (PEG)-plasticized PLA/chitosan/powdered rosemary extract biocomposites of various compositions were characterized in respect to their mechanical and rheological properties, structure by spectroscopy, antioxidant and antimicrobial activities, and in vitro and in vivo biocompatibility. Scanning electron microscopy images evidence the morphology features added by rosemary powder presence in polymeric materials. Incorporation of additives improved elongation at break, antibacterial and antioxidant activity and also biocompatibility. Migration of bioactive components into D1 simulant is slower for PEG-plasticized PLA containing 6 wt % chitosan and 0.5 wt % rosemary extract (PLA/PEG/6CS/0.5 R) biocomposite and it occurred by a diffusion-controlled mechanism. The biocomposites show high hydrophilicity and good in vitro and in vivo biocompatibility. No hematological, biochemical and immunological modifications are induced by subcutaneous implantation of biocomposites. All characteristics of the PEG-plasticized PLA-based biocomposites recommend them as valuable materials for biomedical implants, and as well as for the design of innovative drug delivery systems. Also, the developed biocomposites could be a potential nature-derived active packaging with controlled release of antimicrobial/antioxidant compounds.

## 1. Introduction

The use of natural additives is becoming an increasing interest for the development of new multifunctional materials as a key for new active packaging strategies and new multifunctional biomaterials [1]. Poly (lactic acid) (PLA) selected as the base material in this study has been used in a wide range of applications such as agriculture [2,3], packaging [4], medicine [5,6], etc. Beside its good thermoplasticity and processability, PLA has good biodegradability, biocompatibility, excellent gas barrier properties and mechanical properties. Its high stiffness is usually reduced by the addition of plasticizers [7,8,9,10], which is necessary to get materials adequate for processing, but these additives also lead to a decrease in oxygen barrier and thermal resistance.

The use of fillers that provide a large surface area and interaction with matrix is considered a suitable method to improve mechanical properties, heat resistance, dimensional and thermal stability. Various types of biocomposites with PLA as matrix have been prepared such as: PLA/microcrystalline cellulose [11], PLA/microcrystalline cellulose and silver nanoparticles [12], celulose nanofibers [13], PLA/nanocrystalline cellulose/nanosericite composites [14], hybrid polyethylene glycol (PEG)/graphene oxide [15], sandwiched layers structures [16], etc.

Polymeric matrix serves as a vehicle to incorporate additives (flavor agents, colorants, antioxidants, antimicrobial agents) [17]. Development of new multifunctional materials by the addition of natural additives and/or agricultural waste by-products is an innovative trend and main strategy in polymer science with clear practical interest. The innovative biocomposite formulations containing such additives can accomplish the requirements related to the active packaging functions such as food protection and preservation; offering to packaging new characteristics as improved mechanical, barrier, antioxidant, antimicrobial properties and biological functions; for protecting consumers’ health; marketing; smart communication to consumers; and by their environmental degradation, which will decrease the pollution by a positive environmental impact and reduced waste generation.

Lipids are one of the most chemically unstable food components, which undergo auto-oxidation when foods are exposed to air, light, or metal ions. Their oxidation causes quality deterioration including off-flavor, rancid odor, discoloration, as well as produces some toxic compounds (hydroperoxides, epoxides, oxycholestrols, dimes, etc.), making during time food unacceptable to consumers [18]. Hydroperoxides, as the major initial (auto)oxidation products decompose to secondary compounds such as hexanal, pentanal, and malonaldehyde, which are responsible for off-flavors and odors. Then the acids, alcohols, aldehydes, carbonyls, and ketones are formed which further decompose and/or polymerize resulting in toxic compounds that damage either cells or tissues [19]. Therefore the use of a suitable antioxidant/antibacterial additive is important both for food preservation (in active packaging systems) and to improve the human health (as antimicrobial medical devices).

Antimicrobials can be classified into two major groups [20,21,22]: chemical agents (triclosan benzoates, sorbates, etc.) and natural agents (components of plant extracts and essential oils, bacteriocins and bio-preservatives). Their action is essential in reducing or even eliminating some of the main food spoilage causes, such as rancidity, color loss/change, nutrient losses, dehydration, microbial proliferation, senescence, gas build-up, and off-odors.

Antioxidant and mainly antimicrobial activities [23] both to Gram-negative, Gram-positive bacteria and also fungi are well-recognized functions of the chitosan (CS). It also exhibits outstanding biocompatibility [24] and biodegradability, being a relevant candidate in the field of biomaterials [25]. The presence of amino and hydroxyl groups in the chitosan structure offers along with antimicrobial and antioxidant [26] activities also analgesic [25], mucoadhesive [27], and haemostatic properties [28]. Moreover, it biodegrades into non-toxic residues [29,30]. The amino groups are also able to efficiently make complexes with various species as metal ions or natural/synthetic anionic species such as lipids, proteins, DNA and some negatively charged synthetic polymers as poly(acrylic acid), alginate [31], etc. being also often used to recover heavy metals [32] or for beverages (wine, juices, etc.) clarification [33]. CS was tested in many biomedical [32,34] and pharmaceutical [35] applications, as sutures, dental [36,37] and bone implants [38,39,40,41]. It was approved by the Food and Drug Administration (FDA) for use in wound dressings [42] and food packaging. As already demonstrated, chitosan can inhibit the reactive oxygen species (ROS) by donation of a hydrogen or the one pairs of electrons [43] and prevent the lipid oxidation in food and biological systems [44]. It is known that a combination of two or three main mechanisms can explain chitosan antibacterial and antifungal activities [45]. Briefly, the positively charged chitosan can interact with negatively charged groups at the surface of microbial cells membranes and alter their permeability, this being the first mechanism proposed. The created internal osmotic imbalance leads to inhibit the growth of microorganisms. The second mechanism involves the binding of chitosan with the cell DNA *via* protonated amino groups inhibiting the microbial RNA synthesis. The third mechanism is based on the chelation of metals, suppression of spore elements and binding to essential nutrients to microbial growth [46,47]. It is possible that all such events to occur simultaneously with different intensities. Therefore, CS accomplishes the functions desired for multifunctional materials with various applications. The development of new biodegradable packaging materials, such as PLA/CS films, could be an interesting alternative to petroleum-based synthetic polymers.

Synthetic antioxidants presence in food is questionable due to potential risks and they require strict legislative control. An alternative that is being widely studied is the use of phenolic type natural antioxidants from plant species, including both integral extracts obtained by diverse methods and their purified components (such as catechin, quercetin, caffeic acid, etc.). In recent years, plant extracts have captured the attention of researchers. Spices and herbs are well-known for their antioxidant properties because they contain phytochemicals (flavonoids, tannins, phenolic/polyphenolic compounds, as diterpenes and acids), tocopherols, vitamin C, carotenoids, etc. They also act as antiallergic, anti-bacterial (Gram-negative and Gram-positive species), anti-fungal, anti-inflammatory, antiseptic, antibiotic, and anti-cancer agents [48,49,50]. The action of natural additives as vegetable extracts consists in elimination of the main foodborne spoilage causes, dehydration, microbial proliferation, off-odors, antibrowning, etc. [51].

Plant originated antioxidants have been used in oils or lipid containing foods in order to prevent oxidative deterioration [52]. Naturally occurring compounds in rosemary plant (in all forms such as ground, extracts, and essential oils) [53,54] have been reported to exhibit antioxidant properties greater than butylhydroxyanisole (BHA) and equal butylhydroxytoluene BHT [52]. The rosemary extract has antibacterial, antifungal, antiviral, antimicrobial, anti-inflammatory, astringent, and spasmolytic activity [55] and even anticancer properties. The biological properties of rosemary are attributed to the multiple contributions of its different bioactive compounds and to its phytochemical composition rich in (poly)phenolic compounds, mainly diterpenoids such as carnosic acid and carnosol and also both to the positive contribution of flavonoids and co-presence of flavonoids and diterpenes in the plant. The major compounds contributing for antioxidant activity of rosemary can be categorized in three groups: phenolic acids, phenolic diterpenes, and flavonoids [56,57,58]. Other identified active antioxidant chemicals in rosemary plant [59] are: 1, 8-cineol, α-pinene, β-caryophyllene, β-pinene, β-sitosterol, caffeic acid, camphene, carvacrol, carvone, epirosmanol, γ-terpinene, isorosmanol, limonene, linalool, myrcene, p-cymene rosmadial, rosmarinic acid, terpinen-4-ol, ursolic acid and verbenone [60]. Literature reports either rosmarinic acid, an ester of caffeic acid, and 3,4- dihydroxyphenyllactic acid or/and the phenolic diterpenes carnosol and carnosic acid as the principal antioxidative components of the rosemary extract [60,61].

Chemical structures of some major active compounds in rosemary extracts are presented in Figure 1:

There are studies on the use of vegetable extracts in PLA to confer its multifunctional properties. Three essential oils (EO), namely cinnamon, garlic, and clove have been incorporated in PLA by solvent casting method [62], plasticized (with epoxidized jatropha oil as renewable plasticizer) PLA kenaf biocomposites [63]. In the Valdes et al. review [17] are mentioned antimicrobials/antioxidants as: extracts of blueberry, grape seed extracts, green tea extract, raspberry fruits and pomace, citrus extract, grapefruit seed extracts, the combination of mint extracts or pomegranate peel extracts with chitosan and poly(vinyl alcohol), alginate films containing three natural extracts from rosemary, and Asian and Italian essential oils, lemon extracts, propolis extracts, olive leafs, etc. Both PEG and EOs led to the formation of flexible PLA/PEG/EO films with significant drop in the glass transition temperature (Tg) and mechanical properties. Cinnamon and clove oil–based PLA/PEG films exhibited a complete zone of inhibition against *Campylobacter jejuni* at the maximum concentration whereas the garlic oil–based film had the lowest activity. Improvement in mechanical properties of EO-based films requires serious attention so the films could be processed through melt extrusion process because of loss of EOs at high temperatures.

The depletion of additive is a major problem that determines the quality of materials for long-term usage. Rosemary alcoholic extract in powdered form shows an efficient preservation of initial features in comparison with other phenolic antioxidants [64]. Taking into consideration the above-mentioned specific properties of each of the selected components and a possible synergy of their combination, the purpose of the present study is to develop new multifunctional environmentally friendly materials having applications both in medical and food packaging fields. In our previous paper [65] it was demonstrated that the rosemary alcoholic extract incorporation into PLA confers good elongation at break, rheological properties, antioxidant and antimicrobial activities, barrier properties and a good biocompatibility. A number of studies have been carried out on physical, mechanical, and morphological properties evaluation of PLA/CS system [65,66,67,68,69]. These systems show good antimicrobial properties but they have inferior flexibility and antioxidant properties. However, no single research work has been devoted to the blending of plasticized PLA/CS/rosemary extract (R) and the evaluation of their mechanical and morphological features, antioxidant, antimicrobial, *in vitro* and *in vivo* biocompatibility, and additives migration properties. Motivated by earlier work [65], the current study was carried out to investigate the effect of CS and R on mechanical properties and morphological characteristics of biocomposites obtained by melt mixing and to find the effect of CS/R loading on resulting multifunctional PLA-based biocomposites for food packaging and medical applications, using an innovative combination of CS and R as reinforcements and their antimicrobial, antioxidant activities and biological functions.

## 2. Experimental

### 2.1. Materials

Poly(lactic acid) (PLA) (trade name: PLA 2002D) from Nature Works LLC, UK obtained from renewable resources, transparent, with a melt flow index of 5–7 g/10 min (conditions 210 °C/2.16 kg) and a content of 96% L-lactide and 4% isomer D was used. Average molecular weight determined by GPC was 4475 kDa. According to the literature data it has a density of 1.25 g/cm^3^, melting point of 152 °C, glass transition temperature of 58 °C, the crystallinity depends on isomer content and thermal history, water permeability at 25 °C is 172 g/m^2^ day and percentage of biodegradation/mineralization is 100%.

Chitosan medium molecular (CS) with 200–800 cP viscosity in 1% acetic acid, 75–85% deacetylation degree and MW = 190,000–300,000 g/mol, was provided by Sigma-Aldrich and used as received.

Rosemary ethanol extract (R) in powder form was obtained in Laboratory of Radiation Chemistry, INCDIE - ICPE CA, Bucharest, Romania following a previously reported procedure by the solvent extraction method in a Soxhlet unit [70]. Ethanol was used as an extraction solvent. Rosemary leaves were collected from local farms, dried at ambient temperature and subsequently milled. After collection of the rosemary extract in an ethanol solution, the powder was separated by precipitation induced by the addition of water. The insoluble material was filtered and washed with acetone until it was dried. The extract was further dried under vacuum at ambient temperature. A greenish-yellow fine powder was obtained and stored in desiccators to avoid the absorption of moisture. Its main components are: carnosol, carnosic acid and rosmarinic acid. Its amount of total phenols was determined by Folin-Ciocalteu’s reagent method as described by Scalbert et al. [71]. The resulted total phenolics content was of 112.5 mg GAE (Gallic acid Equivalents)/g dw (dry weight) [65]. Total flavonoid content was measured by the aluminium chloride colorimetric assay [72] and it was of 261.5 (mg Quercetin Equivalents/g dw). The radical scavenging activity of the powdered ethanolic extract of rosemary was determined by ABTS^•+^ (2,2′-azino-bis(3-ethylbenzothiazoline-6-sulfonic acid diammonium radical cation) and a IC_50_ (IC_50_ value is the concentration of the sample required to inhibit 50% of radicals) of 26 µg/mL was found. It was concluded and demonstrated [65] that it exhibits a very good antioxidant and antibacterial activity, biological function and a good biocompatibility to PLA. Present study intended to evidence its effect on PEG-plasticized PLA together with chitosan as the main antibacterial agent and other important biological functions.

Poly(ethylene glycol (PEG) BioUltra 4000 (Sigma-Aldrich, Steinheim, Germany) was used as plasticizer. Its molecular weight is 4000. It is seems that it also show some antimicrobial activity [1] and is crystalizable [73].

### 2.2. Biocomposites Processing

PLA-based biocomposites were prepared using different amounts of chitosan or/and rosemary extract by incorporation them into PLA matrix in melt state. Prior to blend preparation, PLA and the additives were dried in a vacuum oven for 6 h at 80 °C. The processing of PLA/PEG/Rosemary/Chitosan systems was performed at 165 °C for 10 min, at a rotor speed of 60 rpm, using a Brabender station (Brabender^®^ Plasti-Corder^®^ Lab-Station/Lab-Station EC, Brabender GmbH & Co. KG, Duisburg, Germany) due to the improvement of the flow in melt state induced by the presence of the PEG as plasticizer. Specimens for the mechanical characterization were prepared by compression molding using a Carver press with a pre-pressing step of 3 min at 50 atm and a pressing step of 2 min at 150 atm. The compositions of the prepared systems are shown in Table 1.

The optimum amount of 0.5 wt % of R was established in our previous paper [65]. Now the effect of chitosan amount and presence of two bioactive compounds on PLA properties are investigated.

### 2.3. Investigation Methods

#### 2.3.1. Processing Behavior

Processing behavior was evaluated by analysis of processing characteristics following the torque-time curves registered during blending on a Brabender mixer.

#### 2.3.2. Scanning Electron Microscopy (SEM)

The SEM studies were performed on samples fixed on copper supports. The surface was examined by using an Environmental Scanning Electron Microscope (ESEM) type Quanta 200 instrument, (FEI Company, Hillsboro, TX, USA) operating at 25 kV with secondary electrons in low vacuum mode (LFD detector).

#### 2.3.3. ATR–FTIR Spectroscopy

The ATR–FTIR spectra were recorded on a Bruker VERTEX 70 spectrometer (Ettlingen, Germany) with a 4 cm^−1^ resolution. The background and sample spectra were obtained in the 600–4000 cm^−1^ wavenumber range. Spectral processing was achieved with the OPUS program.

#### 2.3.4. Stress-Strain Measurements

The stress-strain measurements were performed at room temperature on dumbbell-shaped samples (1 mm thickness), on an Instron Single Column Systems tensile testing machine (model 3345, Norwood, MA, USA) equipped with a 1kN load cell. The cross-head speed used was of 10 mm/min, and gauge length of 40 mm. Young modulus, tensile strength at break and strain at break have been evaluated according to EN ISO 527-2/2012.

#### 2.3.5. Impact Tests

The unnotched Charpy impact strength of the composites was tested according to EN ISO 179:2000 by means of a CEAST testing machine (Impact Testing Systems—Instron, Pianezza, Italy) with a pendulum of 50 J. Each reported value is the average of at least 5 determinations, both for tensile, as well as for impact evaluation. Before measurements, samples were conditioned 24 h at 23 °C and 50% RH, the same conditions being used for tensile testing.

#### 2.3.6. Dynamic Rheology

The rheological properties of the PLA/R composites containing or not chitosan were measured by means of Anton Paar MCR301 rheometer (MCR301, Graz, Austria) using parallel-plate geometry (diameter of 25 mm). Oscillatory frequency sweeps ranging from 0.05 to 500 rad/s with a fixed strain of 10% (falling in the linear viscoelasticity region) were performed at 165 °C for the composite samples.

#### 2.3.7. Antioxidant Activity Evaluation by ABTS^•+^ (2, 2’-azino-bis 3-ethylbenzthiazoline-6-sulfonic acid) Radical-cation Scavenging Assay

The ABTS^•+^ scavenging test is used to determine the antioxidant activity of both hydrophilic and hydrophobic compounds. The stock solutions included 7 mM ABTS^•+^ solution and 2.4 mM potassium persulfate solution. The working solution was then prepared by mixing the two stock solutions in equal quantities and allowing them to react for 16 h at room temperature in the dark. The solution was then diluted by mixing 1 mL ABTS^•+^ solution with 60 mL ethanol to obtain an absorbance of 0.7–0.8 units at 750 nm using a spectrophotometer (Cary 60 UV-VIS, Agilent Technologies Santa Clara, CA, United States). Fresh ABTS^•+^ solution was prepared for each assay. The reaction between ABTS^•+^ and potassium persulfate directly generates the blue green ABTS^•+^ chromophore, which can be reduced by an antioxidant, thereby resulting in a loss of absorbance at 750 nm. In the case of polymeric biocomposites the reaction mixture consisted of adding 0.4 mL of alcoholic extracts and 2 mL of ABTS^•+^ radical solution and allowed to react for 6 min in a dark environment, and then the absorbance was measured. The alcoholic extract was obtained by placing 80 mg of solid sample in 5 mL ethanol and stirred for 24 h at room temperature (20 °C). The concentration of the alcoholic extracts that produced between 20–80% inhibitions of the blank absorbance was determined and adapted. Scavenging capacity of the composites was compared with that of a standard, namely vitamin E (α-tocopherol), which was used in a concentration of 0.4 mg/mL (2 mg vitamin E in 5 mL ethanol) and the same procedure described above was applied. The antioxidant capacity is expressed as percentage inhibition, calculated using the following formula:(1)$$Inhibition\left(\%\right)=\left[\frac{A_{control}-A_{sample}}{A_{control}}\times 100\right]$$
where *A_control_* is the absorbance of ABTS^•+^ radical in blank and *A_sample_* is the absorbance of an ABTS^•+^ radical solution mixed with extract sample/standard.

#### 2.3.8. Antimicrobial Activity

The experimental protocol for testing antimicrobial efficiency against *Escherichia coli*, *Salmonella. enteritidis* and *Bacillus cereus*, consists in the following stages:

1) Sterilization of the samples, which was performed in autoclave at 110 °C, 0.5 bars for 20 min.

2) Preparation of ATCC cultures was done by: seeding the average pre-enrichment and incubation at 37 °C for 24 h; counting the colonies in 0.1 mL culture by selective culture medium separation; seeding of 0.1 mL bacterial culture ATCC using sterile swab samples surface;

3) Incubation of samples contaminated with the ATCC for 24 h at 25 °C, in the dark, in sterilized glass containers, repeated for other 24 h incubation;

4) Identifying target germs: The following standardized methods of bacteriology procedures were used, according to standards in force: SR ISO 16649- *E. coli*; ISO 7932:2004 ISO 21871:2006(en) - *Bacillus cereus*, SR EN ISO 6579- *Salmonella sp*.

#### 2.3.9. Migration Study of the Active Components within Powdered Rosemary Ethanol Extract into Food Simulant from PLA/PEG/CS-based Films

Double-sided, total immersion migration tests were performed with pieces of films of ~1 cm^2^, soaked in 5 mL of 50% aqueous ethanolic solution, known as a modified D1 food simulant [74] for foods with lipophilic character, considered the most severe aqueous food simulant for alcohol containing food products and milk. [75].

The conditions chosen for migration study simulate storage at ambient temperature for unlimited duration. Since increasing temperature accelerates migration, the samples were kept in an oven at 40 °C for 14 days (minimum 10 days).

At predetermined time intervals, aliquots of ~1 mL were withdrawn from the release medium and were analyzed using a Cary 60 UV-VIS spectrophotometer (Agilent Technologies) by scanning from 200 to 600 nm. The active components content in the food simulant was determined by UV-VIS spectroscopy (detection wavelength λ_max_ = 275 nm). Samples were run in quartz cuvettes with 1 mm path length. Previously a blank test for the simulant and each type of control sample was carried out. The released active components concentrations were calculated based on the calibration curves previously determined for rosemary extract corresponding to the main components of this as rosmarinic acid, carnosol and carnosic acid according to the literature data [76,77]. The corresponding release curves were represented as time dependent plots of the cumulative percentage of active component released.

The migration kinetics parameters were calculated based on the following Equations (2)–(5).

First the data were fitted by the empirical model proposed by Peppas - equation 2 [78,79]:(2)mtm∞=k·tn
where m_t_/m_∞_ represents the fraction of bioactive compound(s) released at time *t*, *n* is the release exponent and *k* is the release rate constant.

The power law release exponent *n* describes the release mechanism from a thin polymer sample: a value of *n* = 0.5 corresponds to Fickian diffusion mechanism, 0.5 < *n* < 1 to non-Fickian/anomalous transport, *n* = 1 to Case II transport, and *n* > 1 to super case II transport [78,79].

Fitting the experimental data to first order kinetics model (Equation (3)):(3)$$ln\left(1\;-\;\frac{m_{t}}{m_{\infty }}\right)=\;-\;k_{1}\cdot t$$
the release rate constant, *k_1_* can be calculated.

For short-term migration, defined as the time for which m_t_/m_∞_ < 0.6, a simplified migration model derived from Fick’s second law is applied, which considers *diffusion* as the main process governing the release of the active component, which occurs from both sides of the film and described by Equation (4):(4)$$\frac{m_{t}}{m_{\infty }}=4{\left(\frac{Dt}{\pi l^{2}}\right)}^{1/2}$$
where *D* is the diffusion coefficient and *l* is the film thickness. A plot of m_t_/m_∞_ versus t^1/2^ should yield a straight line from which the diffusion coefficient can be obtained.

In a two-phase food/polymer system, migrant transfer from one phase to the other one occurs to reach thermodynamic equilibrium. The partition coefficient can be defined as the ratio of the migrant concentration in the film (C_f,∞_) to the migrant concentration in the food simulant system (C_s,∞_) at equilibrium (Equation (5)) [80]:(5)$$K_{p}=\frac{C_{f,\infty }}{C_{s,\infty }}$$
*K_p_* is a measure of the chemical affinity of the migrant towards the film and the food/simulant. When *K_p_* = 1, the migrant concentration in the food simulant system equals the concentration in the film at equilibrium. K_p_ > 1 and K_p_ < 1 describe a higher affinity of the migrant towards the film and respectively a higher affinity of the migrant towards the food system [80].

#### 2.3.10. Biocompatibility Evaluation

a. *In vitro* biocompatibility evaluation - Contact angle (CA) and surface free energy (SFE).

The static contact angle (CA) was determined by the sessile drop method, at room temperature and controlled humidity, within 10 s, after placing 1 μL drop of water on the film surface, using a contact angle goniometer (CAM-200, KSV, Helsinki, Finland). Composite material surfaces may not be completely homogenous, so the surface energies are not evenly distributed; therefore, the measurements of contact angles on solid surfaces were taken at least on 10 points on the tested surface and the average values recorded were used to evaluate the wettability of materials. For more details on the method, see references [81,82]. To obtain the components of the surface free energy (SFE) and the total SFE of the polymer films, the CA at equilibrium between the film surface and three pure liquid, twice-distilled water, formamide and methylene iodide (as purchased at maximum obtainable purity), was measured by fitting the drop profile using the Young-Laplace equation [82,83,84,85]. The total and the components of SFE were calculated by using the Lifshitz-van der Waals acid/base approach of van Oss and Good [86], which divides the total SFE into dispersive Lifshitz-van der Waals interactions ($\gamma_{sv}^{LW}$) and polar Lewis acid-base interactions ($\gamma_{sv}^{AB}$) (Equation (8)). The acid-base interactions are subdivided into electron donor $\gamma_{sv}^{-}$ (Lewis base) and electron acceptor $\gamma_{sv}^{+}$ (Lewis acid) parts (Equations (6)–(8)):(6)$$\left(1+cos\theta \right)\gamma_{lv}=2\left(\sqrt{\gamma_{sv}^{LW}\gamma_{lv}^{LW}}+\sqrt{\gamma_{sv}^{+}\gamma_{lv}^{-}}+\sqrt{\gamma_{sv}^{-}\gamma_{lv}^{+}}\right)$$
(7)$$\gamma_{sv}^{AB}=2\sqrt{\gamma_{sv}^{+}\gamma_{sv}^{-}}$$
(8)$$\gamma_{sv}^{TOT}=\gamma_{sv}^{LW}+\gamma_{sv}^{AB}$$
where θ is the contact angle, $\gamma_{lv}$ is the liquid’s total surface tension, and $\gamma_{lv}^{LW}$ and $\gamma_{sv}^{LW}$ are the apolar (dispersive) Lifshitz–van der Waals components of the liquid and the solid, respectively, whereas $\gamma_{sv}^{+}\gamma_{lv}^{-}$ and $\gamma_{sv}^{-}\gamma_{lv}^{+}$ are the Lewis acid–base contributions of either the solid or the liquid phase. To solve the resulting systems of equations, it is necessary to use at least three test liquids with known $\gamma_{lv}$, $\gamma_{lv}^{LW}$, $\gamma_{lv}^{-}$ and $\gamma_{lv}^{+}$. The subscripts ‘*lv*’ and ‘*sv*’ denote the interfacial liquid-vapor and solid-vapor tensions.

In Table 2, the known surface free energy parameters for the three test liquids, red blood cell membrane, and platelets [87,88,89,90,91] are listed.

Blood compatibility is dictated by the manner in which their surfaces interact with blood constituents, like red blood cells and platelets. The measurement of the surface and interfacial free energy of a material constitute an *in vitro* method for determining the biocompatibility. For establishing material’s compatibility with blood, Equation (9) was used, where Ws/rbc and Ws/p describe the work of spreading of red blood cells and platelets [91]. When blood is exposed to a biomaterial surface, adhesion of cells occurs, and the extent of adhesion decides the life of the implanted biomaterials; thus, cellular adhesion to biomaterial surfaces could activate coagulation and the immunological cascades [92].
(9)$$W_{s}=W_{a}-W_{c}=2\left(\sqrt{\gamma_{sv}^{LW}\cdot \gamma_{lv}^{LW}}+\sqrt{\gamma_{sv}^{+}\cdot \gamma_{lv}^{-}}+\sqrt{\gamma_{sv}^{-}\cdot \gamma_{lv}^{+}}\right)-2\gamma_{lv}$$
where *Ws*—work of spreading (the negative free energy associated with spreading liquid over the solid surface); *Wa*—work of adhesion (defined as the work required separating the liquid and solid phases) and *Wc*—work of cohesion (defined as the work required separating a liquid into two parts). [83]

b. *In vivo* Biological evaluation

White male Wistar rats (200–250 g) were used in the experiment. The animals were housed under a standard laboratory environment (relative humidity 55–65%, chamber temperature 23.0 ± 2.0 °C and 12 h light: dark sequence (lights on at 6:00 a.m.) and fed with a specific diet and water *ad libitum*, excluding the time of the investigations. Before the assessment, the animals were positioned on a raised wire mesh, under a clear plexiglass container and allowed 2 h to familiarize to the testing room. Adult white male Wistar rats weighing 200–250 g from “Grigore T. Popa” University of Medicine and Pharmacy, Iasi, Romania, bio-base were used in the research. They were kept in clean plexiglass cages, at 23.0 ± 2.0 °C constant temperature, relative humidity 55–65% and light/dark cycle of 12/12 h, for 1 week before starting the investigations. Rats were housed in batches of two and given *ad libitum* access to standard pellet diet and water, except during the period of the study.

In this experiment, 42 animals were randomly allocated into 7 groups of 6 animals each as follows:

Group 1 (C) distilled water - used as control; Group 2 - PLA/PEG; Group 3 - PLA/PEG/3CS; Group 4 - PLA/PEG/6CS; Group 5 - PLA/PEG/0.5 R; Group 6 - PLA/PEG/3CS/0.5 R; Group 7 - PLA/PEG/6CS/0.5 R.

At the beginning of the study, rats were anesthetized with 50 mg/kg body weight of ketamine, combined with 10 mg/kg body weight of xylazine, and the tested composite films (weighting 62 mg) were placed subcutaneous, in one side in the dorsal zone, after the skin was shaved and antisepticized with 10% povidone/iodine aqueous solution. Sterile cotton pellet, of 62 mg weight, saturated with distilled water (0.3 mL), was subcutaneous inserted in the animals from the first group. Rats were kept under an aseptic environment for seven consecutive days. The implanted pellets reacted as foreign elements inducing a progressive subacute local inflammatory reaction.

On the 8th day, the animals were anesthetized and the pellets, together with the formed granuloma tissue, were dissected out. First of all the wet pellets were weighed, afterwards, were dried overnight at 60 °C, in an incubator, until a constant weight was recorded.

The granulation tissue formation was calculated after deducing the initial weight of pellets (moment zero—M0) from the post-implantation weight of pellets (after eight days—M1).

We estimated the *in vivo* biocompatibility of biocomposites by quantifying the induced hematological, serum biochemical and immunological modifications [93] by their subcutaneous implantation in rats.

After 24 h, respectively 7 days in the experiment, 0.3 mL of blood samples were obtained from the retro-orbital venous plexus and the following elements were monitored: blood count, hepatic enzymes activity (aspartate transaminase (AST), alanine aminotransferase (ALT), lactic dehydrogenase (LDH), serum urea and creatinine values [94,95]. Serum complement level and the phagocytic capacity of peripheral neutrophils (Nitro Blue Toluene (NBT) test), frequently used to evaluate the influence of pharmacologic agents on the immune defense capacity of laboratory animals, were also assessed [96,97].

Data were presented as mean +/− standard deviation (S.D.). The significance of differences between groups was analyzed using SPSS variant 17.0 for Windows 10 and ANOVA one-way method. The values of p coefficient (probability) below 0.05 are considered as significant versus control.

The experimental protocol was approved and implemented, according to the recommendations of the “Grigore T. Popa” Iasi University Committee for Research and Ethical Issues, concordantly with international ethical normative of the European Directive 2010/63/EU. [98] Each animal was used once and the duration of the experiments was maintained as short as possible. For ethical reasons, all the animals were sacrificed at the end of the study [99].

## 3. Results and Discussion

### 3.1. Processing Behavior

Processing behavior is affected by additives (PEG, CS and R) incorporation at the beginning of mixing. This conclusion results from the comparison of the torque—time curves recorded during processing on Brabender plastograph. The values of the evaluated processing parameters, namely TQ_max_, maximum torque; TQ_1min_, torque after one minute of mixing; TQ_max2_, maximum torque after 1.5 min of mixing dependent on the type of additive and sometimes on their concentration (Table 3). The CS incorporation decreases TQ_max1_ and TQ_1min_ with its increasing concentration, but this decrease only appears in samples containing 3CS in the biocomposites containing both additives. At longer processing times, as TQ_final_ the differences were insignificant. The samples containing rosemary extract present the best melt flow at the end of processing.

### 3.2. SEM Results

Plasticized PLA (PLA/PEG) shows a relatively smooth surface, as seen in Figure 2, with no significant defects and a homogeneous morphology aspect, which is also found in the case of PLA/PEG/0.5R system (see Appendix A) exhibiting a uniform distribution of rosemary powder, with relatively good interfacial adhesion because of relatively strong interaction between components. Chitosan and rosemary powder incorporation led to important changes in morphology, mainly at high concentration (PLA/PEG/6CS/0.5R) when the biocomposites showed an nonhomogeneous morphology with some agglomeration areas because of phase separation, which was very evident at high magnification, as was also the presence of R in the polymeric matrix that determines an uneven surface.

### 3.3. ATR-FTIR Spectroscopy Results

FTIR spectroscopy was used to monitor the absorption band shift in specific regions to determine the interactions between functional groups of the PEG-plasticized PLA and additives CS, and R. The corresponding spectra are shown in Figure 3. The spectra are characterized by 4 main spectral regions: -CH stretching at 3000–2850 cm^−1^, C=O stretching at 1750–1745 cm^−1^, C-H bending at 1500–1400 cm^−1^ and -C-O stretching at 1100–1000 cm^−1^ [100], as seen in Figure 3a. The band at 3398 cm^−1^ corresponds to both -NH_2_ and -OH groups, the band at 2926 cm^−1^ can be attributed to -CH stretching, the absorption band at 1746 cm^−1^ is due to C=O stretching, and is present only in systems containing CS. Additionally, the bending vibrations of the -CH_3_, -NH_2_ groups are observed at 1381 cm^−1^ and 1645 cm^−1^, respectively.

As known from the previous study [65], neat PLA exhibits a relatively sharp band with a maximum at 1749 cm^−1^ assigned to carbonyl C=O stretching and at 1267 cm^−1^ appears the bending vibration of it.

The FTIR spectrum of the PEG-plasticized PLA clearly shows the characteristic absorption bands in the region of 3350–3450 cm^−1^, 2750–3000 cm^−1^, and at 1645 cm^−1^ due to O–H bending and stretching vibration, C–H asymmetric stretching vibration and C=O stretching of ester bonds, respectively [101]. As noticed, the C=O band of PLA shifts to a lower wavenumber in the PLA/PEG spectrum due to the possible strong hydrogen bonding with hydroxyl end-groups of PEG. The characteristic wide absorption band of PEG plasticized-PLA at 2730–2984 cm^−1^, which is mainly due to CH_2_-stretching vibration in carbonyl compounds, shows multiple overlapped bands with narrow band intensity upon incorporation of CS. It was also found that the band intensity at 2946 cm^−1^ decreased by the addition of CS. In the FTIR spectrum of PLA/PEG/CS sample, the band at 1752 cm^−1^ assigned to carbonyl stretching vibration is shifted to 1747 cm^−1^ in the spectra of the PLA/PEG/CS/R biocomposites. Therefore, it can be implied that CS is dispersed in the PLA matrix with some levels of interaction between them forming the PLA-CS composites. CS presents a band near 1650 cm^−1^ for the amide carbonyl (overlapped). Similar result has also been reported by other researchers [102]. A significant difference between samples was observed in 1000–1150 cm^−1^ region - Figure 3b - both in bands shape and positions. CS incorporation leads to narrower 1050–1140 cm^−1^ band with higher intensity and the band peak is placed at higher wavenumbers. These shifts in the absorption bands indicate the miscibility and interaction of PLA with additives. A small amount of hydroxyl group (O–H) (band at 3400 cm^−1^) in the biocomposites could be attributed to the chitosan and/or terminal hydroxyl groups in the PLA main chain. The presence of bands at 1448 cm^−1^ for C–H stretching in the CH_3_ and 950 cm^−1^ and 847 cm^−1^ for C–C single bond are assigned to PLA [103]. All these spectral modifications pointed towards good dispersion and interaction between PLA, R, and CS, which significantly changed the morphological characteristics of the composites as appears from SEM images and also properties of the complex material. Interaction between components is also proved by the variation of the surface properties of biocomposites (as seen below).

### 3.4. Mechanical Properties

The results on the variation of the tensile properties of the PLA-based biocomposites in dependence on CS content are given in Figure 4. It is clear that the increase of CS amount in the plasticized PLA increases Young modulus and decreases both tensile strength—Figure 4a—and elongation at break—Figure 4b. Doubling the amount of CS from 3 wt% to 6 wt% increased the Young modulus with 20%. The decrease of the tensile strength and elongation at break for these compositions is not so significant. In the presence of 0.5 wt% R the evolution of the mechanical properties with CS content is totally different. The PLA/PEG/0.5R shows a slight high values for Young modulus and tensile strength but adding CS in this composition led to a PLA/PEG/6CS/05R biocomposite with the highest elongation at break of 52% because of a synergistic effect of these two bioactive compounds in the ratio of 6CS/0.5R. Similar results were found by other authors [104]. In that case, the tensile strength increased up to 5 wt % CS loading and Young’s modulus increased up to 10 wt% with the addition of CS into the matrix, while the percent elongation at break decreased. However, when the CS content was increased to 15 wt%, the tensile strength and tensile modulus were slightly decreased. They attributed these improvements to a good dispersion of CS and attractive interactions between the composites components. The literature gives different results concerning the variation of the mechanical properties including impact ones of PLA by plasticization with PEG. Bijarimi et al. [105] found that tensile and flexural strength, stiffness and notched Izod impact strength decreased significantly when the PEG at 2.5–10 wt% concentrations were added to the PLA matrix It was found that the PLA/chitosan composite materials showed appropriate porosity and structure, and could keep certain shape and mechanical properties [67]. The processing morphological, structural, thermal and mechanical performance of electrospun biocomposites based on PLA blended with 25 wt% of poly(hydroxybutyrate) (PHB), plasticized with 15 wt% of acetyl(tributyl citrate) (ATBC) and loaded with 1–5 wt% of chitosan or catechin microparticles have been studied by Arrieta et al. [106]. Both fillers present a high content of hydroxyl groups on their surfaces and there are interactions between PLA, PHB and plasticizer. Chitosan creates bead defects in the fibers, which leads to a reduction of the mechanical performance of biocomposites, while catechin antioxidant effect improved the thermal stability of biocomposites and produced beads-free fibers with better mechanical performance. Other authors reported that the incorporation of CS particles into PLA led to less rigid and less stretchable films.

### 3.5. Impact Properties

Data related with impact properties of the plasticized PLA-based biocomposites are summarized in Table 4. Both impact strength and impact energy increased with increasing CS content and also by incorporation of alcoholic rosemary extract. Synergistic effect appears in the case of PLA/PEG/3CS/0.5R biocomposite.

Anuar et al. [107] and Ghalia et al. [108] reported that impact strength of a plasticized biocomposite is significantly improved, as compared to unplasticized biocomposite. The explanation is found in the effect of PEG concentration and molecular weight [107]. The results revealed that PEG with high molecular weight and increased concentration up to 20 wt% significantly improves the crystallization capacity and impact toughness of PLA. With increasing the average molecular weight of PEG, the crystallinity and impact strength of PLA/PEG blends first decreased and then increased.

### 3.6. Rheological Properties

Plasticized PLA-based biocomposites predominantly show a viscous behaviour (G″ > G′) in the entire angular frequency region studied, as seen in Figure 5a. Storage and loss moduli dependence on deformation frequency presents the same trend in the variation of the values of the studied biocomposites, more obvious differences being observed at low frequencies and in G′ values. The PLA/PEG/R biocomposite shows the lowest G′ values. Storage modulus G′ after CS incorporation is higher and increases with CS content both in respect with PLA/PEG and PLA/PEG/R. Complex viscosity, as presented in Figure 5b shows a significant increase for PLA/PEG/CS with increasing CS content. The flow curve of the PLA/PEG is the lowest one due to the PEG plasticizer effect. No cross-over point was found for the plasticized PLA-based biocomposites.

It is worth mentioning that a high increase of the viscosity was recorded when rosemary powder was incorporated in the PLA/PEG blend, because some interactions appear between the functional groups of the components. The presence of chitosan in the PLA/PEG system increased the values of the parameters of the melt rheology recorded, a rigidity of the obtained samples being observed also in melt state. On contrary, the loading with chitosan of the PLA/PEG/0.5R system significantly decreased the dynamic viscosity. The Newtonian plateau has been extended to high frequencies for the blends containing rosemary extract and chitosan. The obtained results are in accordance with those found by other authors [109,110].

### 3.7. Antioxidant and Antibacterial Properties

#### 3.7.1. Antioxidant Activity Evaluation

From Figure 6, it is observed that the addition of chitosan and powdered rosemary extract imparts antioxidant property to PEG plasticized PLA matrix.

The ABTS^•+^ radical inhibition activity increases by increasing the content of chitosan and rosemary extract. It is at least two times higher than that of PLA [65] and PLA/PEG by CS incorporation and 10 or 30 times higher when rosemary extract or both active natural compounds are present in formulation. Moreover, a synergism between chitosan and rosemary extract—the strongest antioxidant activity being obtained for the sample PLA/PEG/6CS/0.5R, comparable with the antioxidant activity exerted by vitamin E, is noticed. This research revealed that rosemary ethanolic extract enhances the antiradical efficiency and the antibacterial activity (see below) of chitosan through synergistic interactions. It is interesting to notice that the chitosan addition into the composites matrix improves the antiradical effect. The PLA/PEG/6CS/0.5R sample reveals the highest antioxidant activity. It is known that chitosan-based systems show poor stability, which restricts its practical applicability. It has become a great challenge to establish sufficient shelf-life for chitosan formulations. Improved stability can be assessed by controlling environmental factors, manipulating processing conditions (e.g., temperature), introducing a proper stabilizing compound, developing chitosan blends with another polymer, or modifying the chitosan structure using chemical or ionic agents [111]. Houlihan and others [112] certified in previous studies that some components of rosemary extracts such as rosmanol, carnosol, rosmarinic acid, and carnosic acid can become four times more effective than BHA and BHT in *in vitro* conditions. From the presented results it is seems that R is an effective agent for antioxidative stabilization both of PLA and CS.

#### 3.7.2. Antibacterial Activity Evaluation

Antimicrobial properties of the studied biocomposites were tested for three bacteria: *Bacillus cereus* ATCC 14579 (Gram-positive), *Salmonella typhymurium* ATCC 14028 (Gram-negative bacteria) and *Escherichia coli* ATCC 25922 (Gram-negative)—Table 5. The CS incorporation is very effective at both concentrations used, while adding rosemary alcoholic extract leads to greater values of percentage inhibition for all tested bacteria. A lower activity was found against *Salmonella typhymurium*, however a 100% inhibition against all three bacteria after 48 h was recorded.

Similar results were reported by other authors. PLA/chitosan fibrous membrane exhibited excellent antibacterial capabilities with an inhibition of 99.4% and 99.5% against *E. coli* and *S. aureus*, respectively [113].

PLA/CS composite showed significant antimicrobial activity against total aerobic and coliform microorganisms, especially when the particle size of CS was reduced [69]. Carnosic acid and carnosol as the major constituents of rosemary extract are found to have high antibacterial activity against various microorganisms including *Streptococcus mutans*, *S. salivarius*, *S. sobrinus*, *S. mitis*, *S. sanguinis*, and *Enterococcus faecalis* which initiates dental caries [114]. Rosemary extract has also been found to be promising as a nutritional strategy for improving meat quality.

### 3.8. Migration Study of the Active Components from Powdered Rosemary Alcoholic Extract into Food Simulant from PLA/PEG/CS/R-based Biocomposite Films

In order to obtain the migration profiles corresponding to the investigated samples, the content of active components from rosemary powder in the food simulant was determined by UV-VIS spectroscopy. The registered UV-VIS spectra for the PLA/PEG and PLA/PEG/CS samples containing the rosemary extract showed an absorption maximum at 275 nm, which is shifted with ~10 nm compared with the maxima on the calibration curve and the one observed in our previous investigations on PLA/R samples, at 285 nm [65]. This hypsochromic shift may occur in the absorption spectrum of molecules that contain more chromophores due to interaction of the chromophores, chemical interactions (dissociation or reaction with the solvent) and may give information about the intermolecular structural changes and/or perturbation of the electronic states due to the environmental factors [115].

The UV-VIS absorption spectra of the samples under investigation were influenced by their complex composition (the presence of both PEG and CS), with a higher number of chromophore groups responsible for the absorption, not only the aromatic C=C from the rosemary active compounds, but also the ester and amide C=O groups, taking into consideration that the CS content increases in the composition of these films, while the R content is the same (0.5 wt% R). The release curves of the active constituents of powdered rosemary alcoholic extract by migration into 50% ethanol solution at 40 °C, as food simulant medium, from PLA/PEG/CS-based films prepared by melt mixing are represented in Figure 7.

The release profiles of the active components of the R describing migration phenomenon into selected food simulant show an overall similar release behavior from studied films. Samples PLA/PEG/0.5R and PLA/PEG/3CS/0.5R show a fast release in the first 30 h, then after 70 h the equilibrium is reached in the case of migration from plasticized PLA and from the biocomposite containing 3 wt% CS. At a content of 6 wt% CS the migration is much slower and a more gradual release can be noticed for entire studied period and the equilibrium will be reached after 5 times longer period of about 350 h, although in higher quantity until the end of the time interval of 14 days, an amount of ~ 61% is released compared with 47% from the plasticized PLA/PEG/0.5R sample and 51% from PLA/PEG/3CS/0.5R sample. Thus the increase of CS content in the samples composition favors a gradual release without reaching a plateau even up to 10–12 days.

The calculated kinetic parameters and the corresponding correlation coefficient values (R^2^) are summarized in Table 6.

The *n* and *k* kinetic parameters values obtained by fitting the data with equation 2 indicate a migration behavior with tendency towards Fickian diffusion, especially for sample PLA/PEG/6CS/0.5R (n = 0.38) as suggested also by the release profiles (Figure 7). The release rate constant values, *k*, show a slower release on the first interval (up to 120 h) also for sample PLA/PEG/6CS/0.5R, with the lowest k = 72.92 h^−n^.

By fitting the experimental data to equation 3, significantly lower values of the correlation coefficient (R^2^) were obtained, showing that the first order kinetic model is not the most appropriate to describe the migration process from these samples. The calculated k_1_ values are similar for samples PLA/PEG/0.5R and PLA/PEG/3CS/0.5R and the lowest value was obtained for PLA/PEG/6CS/0.5R, highlighting the slowest release mechanism for this sample, as evidenced by power law model. A good linear fit was obtained for all samples by fitting the data to Equation (4), with correlation coefficients (R^2^ values) ranging between 0.97–0.99, suggesting that experimental data are well described by the diffusion model for short-range times.

It can be seen, that the best fitting of equation 4 corresponds to sample PLA/PEG/6CS/0.5R, which also shows the lowest D value (1.05 × 10^−13^ m^2^/s), taking into consideration that kinetic calculations are based on the first interval of release, for a short-term migration.

The partition coefficient values, K_P_ show that up to 14 days studied interval, for samples PLA/PEG/0.5R and PLA/PEG/3CS/0.5R the migrant concentration in the food simulant system equals the concentration in the film (K_P_ ~ 1)—which can be explained by their similar behavior, with ~ 50% R released. For PLA/PEG/6CS/0.5R sample the K_P_ = 0.64 value (K_p_ < 1) corresponds to a higher affinity of the migrant towards food system, which is preferred in functional films with “positive migration” where additives incorporated into the film are expected to provide a controlled release into the food system for shelf life extension [80,116].

The overall migration levels obtained were of 19.2 mg/kg for PLA/PEG/0.5R, 14.1 mg/kg for PLA/PEG/3CS/0.5R and 18.4 mg/kg for PLA/PEG/6CS/0.5R. It can be noticed that all the values are much lower than the overall migration limits for food contact materials of 60 mg/kg of food simulant, established by the current legislation for food packaging materials in both non-polar and polar simulants [74,117,118].

### 3.9. Biocompatibility Study

#### 3.9.1. *In Vitro* Biocompatibility Evaluation by Contact Angle Measurements—Determination of Wettability, Surface Free Energy, and Work of Spreading

Wettability of the composites was evaluated by the contact angle measurements. In Figure 8 are presented the results obtained for the plasticized PLA biocomposites with or without powdered rosemary ethanolic extract containing different ratios of chitosan.

Chitosan has a polar nature and by adding it to hydrophobic materials such as PLA produces increased surface free energy, which leads to decreased contact angle with water, as revealed in Figure 8. For both sets of samples with or without rosemary ethanolic extract (R) containing chitosan was noticed that the water contact angle values decreased with the increase in chitosan content, which indicates the increase of hydrophilicity of the material. In contrast, in the case of total free surface energy ($\gamma_{sv}^{TOT}$) is observed that it increases proportional with the chitosan content (Table 7) for the samples without R and it decreases for the samples containing R.

This may be due to the presence of an excess of polar groups from chitosan on the surface of the composites without R and by adding R in the PLA/PEG/CS composites some polar groups from chitosan are involved in different interactions with complementary groups from rosemary extract (such as carboxyl, hydroxyl, carbonyl groups) hence no longer available at the composite surface (solid-air interface). This behavior is evidenced also by variation of the polar component of SFE ($\gamma_{sv}^{AB}$) which increases proportional with the chitosan content in the samples without rosemary ethanolic extract, the major contribution to it being brought by the Lewis base component ($\gamma_{sv}^{-}$, electron donor groups such as -OH, -NH_2_), as revealed by the data listed in Table 7. For all the samples, the $\gamma_{sv}^{-}$ parameter is higher than $\gamma_{sv}^{+}$ indicating monopolar behavior, more specifically, a Lewis base character.

Work of spreading for red blood cells (*W_s/rbc_*) has positive values except for PLA/PEG/3CS/0.5R sample and work of spreading for platelets (*W_s/p_*) has negative values. This means that when PLA/CS/R samples come in contact with blood, they cause an increase in the work of cohesion for platelets, therefore the aforementioned blood cells will not adhere easily onto surface of biomaterial, avoiding occurrence of thrombosis, which is an unwanted event for blood-contacting applications. [119,120]

#### 3.9.2. *In Vivo* Biocompatibility

PLA surgical sutures are a new type of absorbable sutures that can be degraded and absorbed in the body. High hydrophobicity of the PLA sutures surface leads to poor biocompatibility and cellular affinity. Enhanced surface hydrophilicity of polylactic acid sutures treated by lipase and chitosan is reported in literature [121]. The sutures were etched by lipase and then grafted with chitosan. Chitosan significantly improved the hydrophilicity of PLA-based materials. This improvement is very important for medical applications as sutures.

In this study none of the animals died during the seven days surveillance period after the pellets implantation. No behavioral changes, such as the decrease in food uptake and lethargy, were observed, in rats with the biocomposite implants.

The subcutaneous implantation of PLA/PEG, PLA/PEG/3CS, PLA/PEG/3CS/0.5R, PLA/PEG/6CS, PLA/PEG/6CS/0.5R films, did not considerable influence the animal’s body weight (Table 8), nor the granuloma’s tissue weight (Figure 9), compared to the control group, after seven days in the experiment.

Comparative analysis between the effects of the studied groups, revealed a slight tendency of PLA/PEG/3CS/0.5R and PLA/PEG/3CS to reduce the animal’s weight, as well as to decrease the granuloma’s tissue weight, but statistically insignificant than those of R, PLA (presented in a previous paper [65] and control groups, during the test (Table 8). The effects of PLA/PEG/3CS/0.5 R were more intense than of PLA/PEG/3CS, probably due to the presence of R incorporated into PLA matrix. The accentuated actions of PLA/PEG/3CS/0.5R compared with those of PLA/PEG/6CS/0.5 R, can be attributed to the different concentrations of CS, but also to the particular ways of releasing the active compounds from the existing matrix.

No substantial changes in the percentage values of polymorphonuclear neutrophils (PMN), lymphocytes (Ly), eosinophils (E), basophils (B) and monocytes (M) between R, PLA, (see ref. [65]) PLA/PEG/3CS, PLA/PEG/3CS/0.5R, PLA/PEG/6CS, PLA/PEG/6CS/0.5R groups and control group, were noted after 24 h, nor the 7^th^ day (Table 9).

Laboratory analysis did not identify significant alterations of AST, ALT and LDH activity, between groups with biocomposite/plasticized PLA implants, and control, after 24 h and seven days the tested substances administration (see Appendix A).

The use of PLA/PEG/3CS, PLA/PEG/3CS/0.5R, PLA/PEG/6CS, PLA/PEG/6CS/0.5 R implants was not accompanied by significant variations of the blood urea and creatinine values compared to the groups treated with control, R, respectively with PLA (see Reference [65] and Appendix A).

The implantation of PLA/PEG/3CS, PLA/PEG/3CS/0.5R, PLA/PEG/6CS, PLA/PEG/6CS/0.5R films did not induce noticeable changes in the serum complement levels and the phagocytic capacity of peripheral neutrophils, compared to plasticized PLA and control group, after 24 h, as well as seven days in the experiment (see Appendix A).

## 4. Conclusions

New plasticized PLA-based multifunctional materials containing additives derived from natural resources were obtained by melt mixing. They show satisfactory mechanical properties and also superior thermal properties (as it will be presented in the next paper).

The developed powdered rosemary ethanol extract/chitosan-incorporated into plasticized polylactide films exhibited good flexibility, antioxidant and antimicrobial activity against both Gram-negative and Gram-positive bacteria. Migration of bioactive components into D1 food simulant is slower for PLA/PEG/6CS/0.5 R biocomposites and occurred by a diffusion controlled mechanism. The biocomposites show a high hydrophilicity and good *in vitro* and *in vivo* biocompatibility. When PLA/CS/R samples come in contact with blood they cause an increase in the work of cohesion for platelets, therefore the aforementioned blood cells will not adhere easily onto surface of biomaterial, avoiding occurrence of thrombosis, which is an unwanted event for blood-contacting applications.

In our experimental conditions, the subcutaneous implantation of biocomposite films induced hematological, biochemical and immunological modifications comparable with control group. The use of implants based on PLA/PEG, PLA/PEG/3CS, PLA/PEG/6CS, PLA/PEG/0.5R, PLA/PEG/3CS/0.5R and PLA/PEG/6CS/0.5R compositions showed a good *in vivo* biocompatibility in rats, thus indicates that, these films represent valuable materials for biomedical implants, and also for the design of innovative drug delivery systems. Also, the developed biocomposites could be a potential nature-derived active packaging with controlled release of antimicrobial compounds, and such films could potentially be used in retail food packaging to control pathogens commonly associated with various foods, such as poultry meat.

Although the studied biocomposites show good features required for biomedical and food-packaging fields, they are not recommended for long-term applications because of the PLA degradability (however it is well-known that the degradability is an advantage for many applications, see above). In addition, a strong acidic environment could limit them because of increased solubility of the chitosan in such conditions.

## Figures and Tables

**Figure 1 polymers-11-00941-f001:**
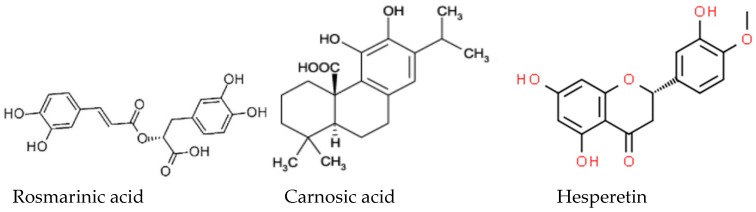
Chemical structure of the major antioxidative compounds in rosemary extracts.

**Figure 2 polymers-11-00941-f002:**
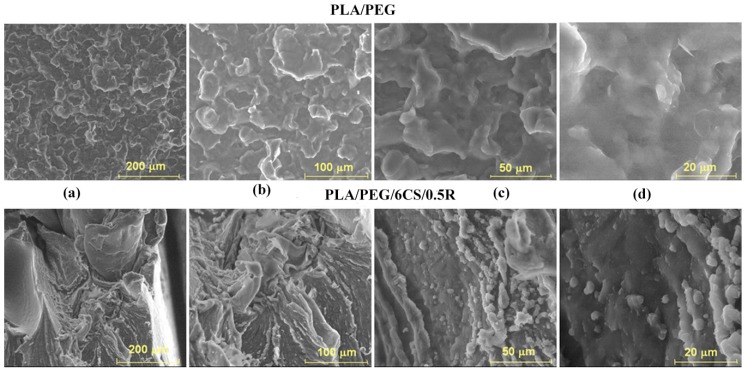
SEM images of poly(lactic acid) (PLA)/polyethylene glycol (PEG) and PLA/PEG/chitosan (CS)/rosemary extract (R) samples at different magnifications: (**a**) 200 µm; (**b**) 100 µm; (**c**) 50 µm and (**d**) 20 µm indicated on figures.

**Figure 3 polymers-11-00941-f003:**
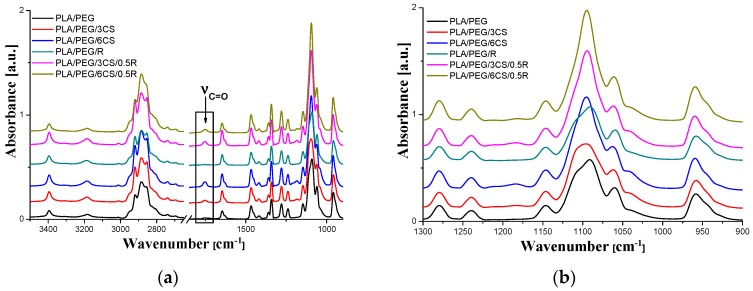
FT-IR results of the plasticized PLA, PLA/PEG/3CS and 6CS, PLA/R, and PLA/PEG/CS/R: (**a**) entire wavenumber region (3500–600 cm^−1^); and (**b**) 1000–1300 cm^−1^.

**Figure 4 polymers-11-00941-f004:**
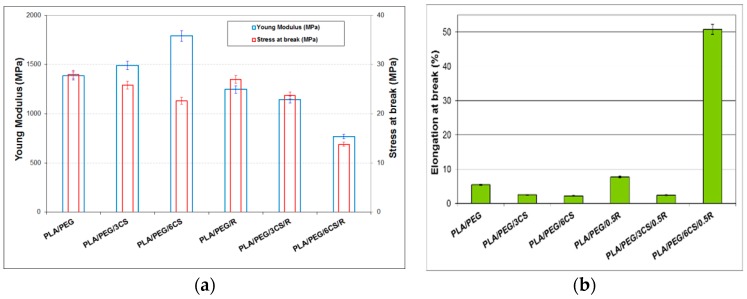
Tensile properties of the PEG-plasticized PLA/CS/R biocomposites: (**a**) Young’s modulus and tensile strength; (**b**) elongation at break.

**Figure 5 polymers-11-00941-f005:**
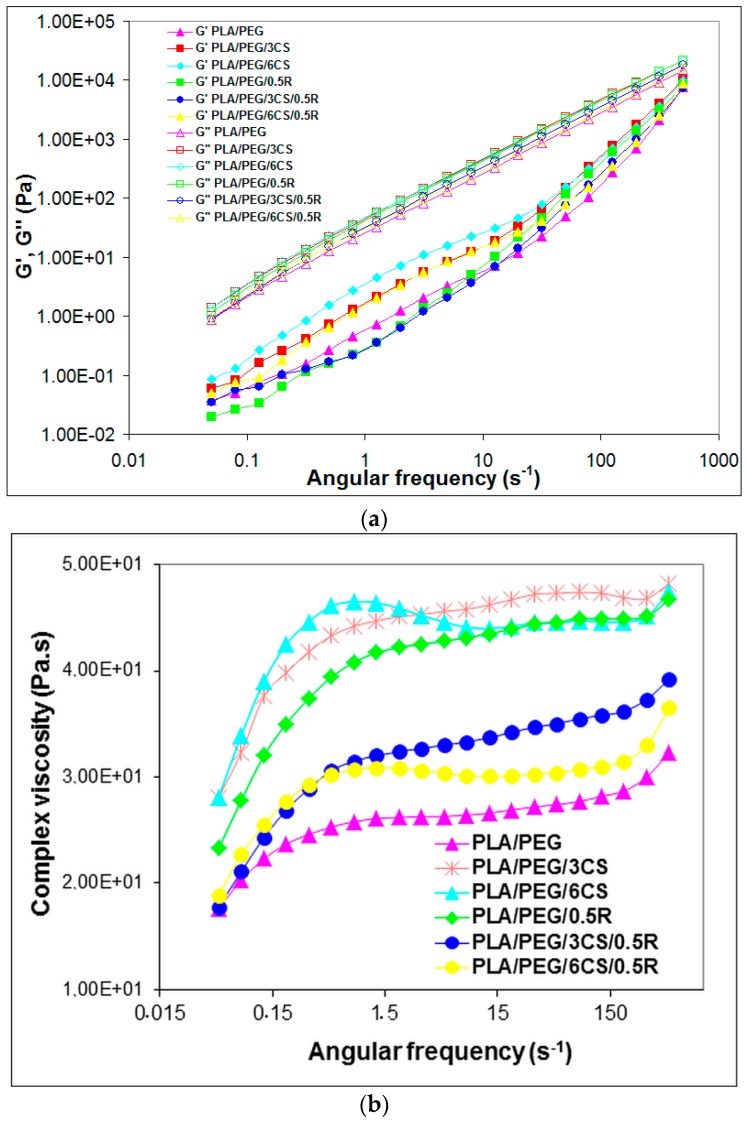
Dependence of the storage modulus (G′) and loss modulus (G″) (**a**), and complex viscosity (**b**) on angular frequency for PLA/PEG and PLA/PEG/CS and PLA/PEG/CS/R systems.

**Figure 6 polymers-11-00941-f006:**
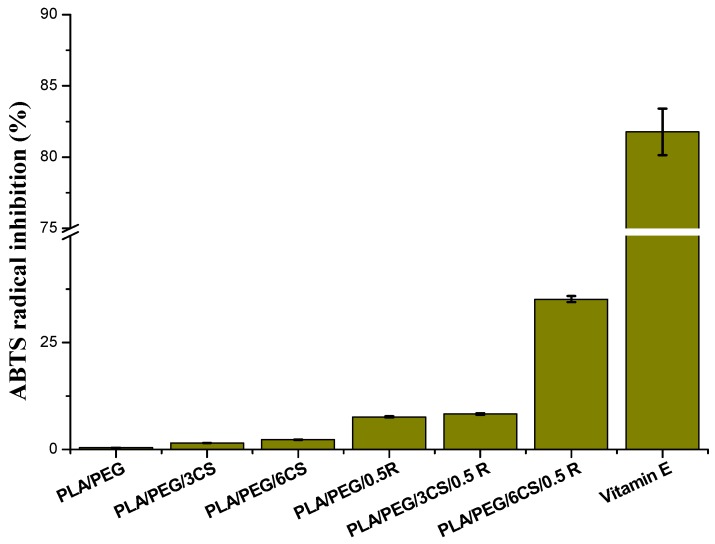
ABTS^•+^ radical inhibition activity of the PEG-plasticized PLA containing chitosan and powdered rosemary ethanolic extract.

**Figure 7 polymers-11-00941-f007:**
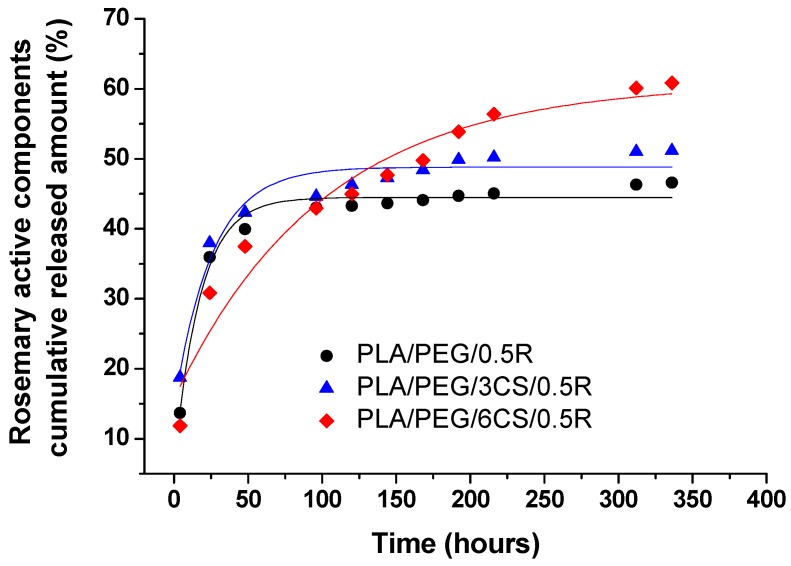
Release profiles of the active components from rosemary powdered ethanol extract into 50% ethanol solution as a food simulant.

**Figure 8 polymers-11-00941-f008:**
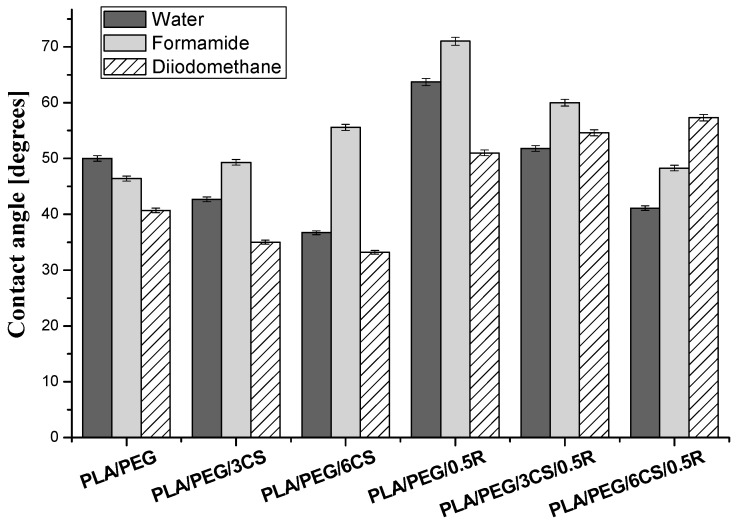
Contact angle variation for plasticized PLA, PLA/CS biocomposites with or without rosemary ethanolic extract.

**Figure 9 polymers-11-00941-f009:**
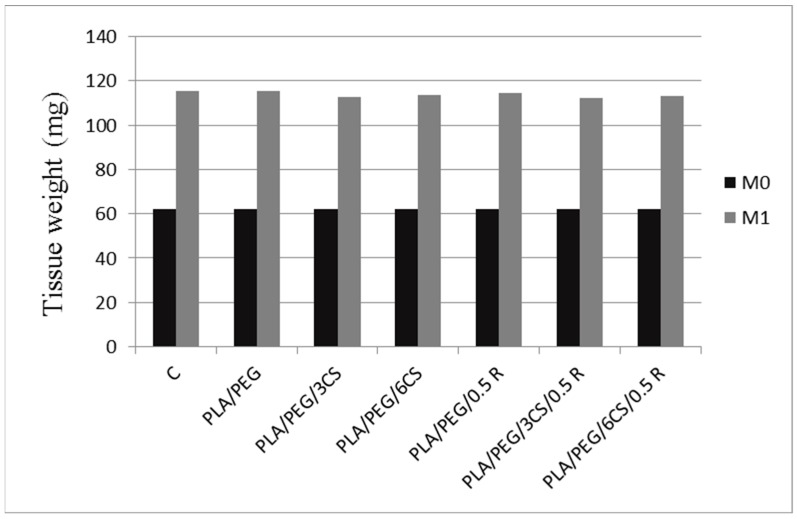
The influence of biocomposite implants on the granuloma’s tissue weight after seven days.

**Table 1 polymers-11-00941-t001:** Designation and compositions of the prepared poly(lactic acid) (PLA)-based biocomposites.

No.	Sample	PLA (wt %)	PEG (wt %)	Chitosan (CS) (wt %)	Rosemary Ethanolic Extract (R) (wt %)
1	PLA/PEG	80	20	-	-
2	PLA/PEG/3CS	77	20	3	-
3	PLA/PEG/6CS	74	20	6	-
4	PLA/PEG/0.5R	79.5	20	-	0.5
5	PLA/PEG/3CS/0.5R	76.5	20	3	0.5
6	PLA/PEG/6CS/0.5R	73.5	20	6	0.5

**Table 2 polymers-11-00941-t002:** Surface free energy parameters (mN/m) of the liquids used for contact angle measurements [87,88,89,90,91].

Liquid	$\gamma_{lv}^{TOT}$	$\gamma_{lv}^{LW}$	$\gamma_{lv}^{AB}$	$\gamma_{lv}^{+}$	$\gamma_{lv}^{-}$
Water	72.80	21.80	51.00	25.50	25.50
Formamide	58.00	39.00	19.00	2.28	39.6
Methylene iodide	50.80	50.80	0.00	0.72	0.00
Red blood cells (rbc)	36.56	35.2	1.36	0.01	46.2
Platelets (p)	118.24	99.14	19.1	12.26	7.44

**Table 3 polymers-11-00941-t003:** Melt processing characteristics of PEG-plasticized PLA and its composites.

Sample	TQ_max1_ (Nm)	TQ_1min_ (Nm)	TQ_max2_ (Nm)	TQ_5min_ (Nm)	TQ_final_ (Nm)
PLA/PEG	12.9	0.9	-	7.3	6.5
PLA/PEG/3CS	8.8	1.7	11.0	6.5	6.0
PLA/PEG/6CS	6.6	1.4	12.1	7.6	6.2
PLA/PEG/0.5R	10.1	2.8	-	7.3	5.7
PLA/PEG/3CS/0.5R	8.7	1.5	10.6	6.1	5.9
PLA/PEG/6CS/0.5R	9.5	2.8	12.6	7.1	5.5

TQ_max1_, maximum torque; TQ_1min_, torque after one minute of mixing; TQ_max2_, maximum torque after 1.5 min of mixing; TQ_5min_, torque after 5 min of mixing (half processing time); TQ_final_, torque at the end of mixing.

**Table 4 polymers-11-00941-t004:** Impact properties of the plasticized PLA-based biocomposites containing chitosan and alcoholic rosemary extract.

Sample	Impact Strength (kJ/m^2^)	Impact Energy (J)
PLA/PEG	2.21	0.11
PLA/PEG/3CS	3.18	0.13
PLA/PEG/6CS	21.15	0.88
PLA/PEG/0.5R	4.52	0.18
PLA/PEG/3CS/0.5R	15.20	0.62
PLA/PEG/6CS/0.5R	12.39	0.51

**Table 5 polymers-11-00941-t005:** Antibacterial activity of the chitosan and rosemary alcoholic extract incorporated into PLA-based materials against *Bacillus cereus, Salmonella typhymurium* and *Escherichia coli*.

Sample	*Inhibition (%)*					
	*Bacillus cereus*		*Salmonella typhymurium*		*Escherichia coli*	
	24 h	48 h	24 h	48 h	24 h	48 h
LDPE	6	18	3	8	7	13
PLA/PEG	45	91	29	77	69	94
PLA/PEG/3CS	86	100	58	100	73	100
PLA/PEG/6CS	86	100	58	100	73	100
PLA/PEG/0.5R	86	100	48	100	76	100
PLA/PEG/3CS/0.5R	100	100	81	100	76	100
PLA/PEG/6CS/0.5R	100	100	90	100	82	100

**Table 6 polymers-11-00941-t006:** Kinetic parameters of the bioactive compounds release from plasticized PLA based materials containing CS and rosemary extract.

Samples	Peppas/Power Law Model				First Order Kinetic Model		Diffusion Model		K_P_
	n	R^2^	k × 10^3^ (h^−n^)	R^2^	k_1_ × 10^3^ (h^−n^)	R^2^	D × 10^−13^ (m^2^/s)	R^2^	
PLA/PEG/0.5R	0.37	0.99	85.12	0.98	5.20	0.84	1.7	0.98	1.06
PLA/PEG/3CS/0.5R	0.23	0.98	147.78	0.99	5.17	0.81	2.05	0.97	0.95
PLA/PEG/6CS/0.5R	0.38	0.99	72.92	0.99	3.7	0.92	1.05	0.99	0.64

**Table 7 polymers-11-00941-t007:** Surface free energy (mN/m) of PLA/PEG-based samples containing chitosan and powdered rosemary ethanolic extract and work of spreading (*W_s_*) for red blood cells (rbc) and platelets (p).

Sample Code	$\gamma_{sv}^{LW}$	$\gamma_{sv}^{AB}$	$\gamma_{sv}^{+}$	$\gamma_{sv}^{-}$	$\gamma_{sv}^{TOT}$	*W_s/rbc_*	*W_s/p_*
PLA/PEG	39.17	1.92	0.03	34.91	41.09	4.54	−69.58
PLA/PEG/3CS	41.94	6.98	0.25	49.47	48.92	14.42	−49.46
PLA/PEG/6CS	42.76	20.34	1.54	67.30	63.10	22.22	−43.83
PLA/PEG/0.5R	33.64	13.95	1.27	38.25	47.59	12.27	−71.51
PLA/PEG/3CS/0.5R	31.61	3.90	0.08	45.87	35.51	−1.14	−75.52
PLA/PEG/6CS/0.5R	30.07	7.72	0.30	49.53	37.78	0.81	−75.01

$\gamma_{sv}^{TOT}$—total surface tension; $\gamma_{sv}^{LW}$—apolar Lifshitz-van der Waals component; $\gamma_{sv}^{AB}$—polar Lewis acid-base interaction; $\gamma_{sv}^{+}$—electron acceptor (Lewis acid) component; $\gamma_{sv}^{-}$—electron donor (Lewis base) component; W_s/rbc_ and W_s/p_—work of spreading for red blood cells and platelets, respectively.

**Table 8 polymers-11-00941-t008:** The influence of biocomposite implants on the animals body weight after seven days.

Groups	Variation of the Mean Animal’s Weight (g)
**C**	+8.2
**PLA/PEG**	+8.1
**PLA/PEG/3CS**	+7.6
**PLA/PEG/6CS**	+7.9
**PLA/PEG/0.5R**	+8.0
**PLA/PEG/3CS/0.5R**	+7.3
**PLA/PEG/6CS/0.5R**	+7.7

**Table 9 polymers-11-00941-t009:** The influence of biocomposite implants administration on the elements of the leucocyte formula. Values were presented as mean ± S.D. for six animals in a group.

Groups		Leucocyte Formula (% Values)				
		PMN	Ly	E	M	B
Control	24 h	29.5 ± 0.83	66.3 ± 2.11	0.6 ± 0.08	3.4 ± 0.10	0.2 ± 0.10
	7 days	29.7 ± 0.47	65.9 ± 1.93	0.7 ± 0.10	3.5 ± 0.10	0.2 ± 0.05
PLA/PEG	24 h	29.8 ± 0.89	65.7 ± 1.39	0.7 ± 0.10	3.6 ± 0.05	0.2 ± 0.04
	7 days	29.9 ± 0.55	65.5 ± 1.63	0.7 ± 0.05	3.7 ± 0.12	0.2 ± 0.05
PLA/PEG/3CS	24 h	29.1 ± 0.37	66.7 ± 2.14	0.6 ± 0.08	3.4 ± 0.05	0.2 ± 0.05
	7 days	29.4 ± 0.98	66.1 ± 1.33	0.8 ± 0.1	3.5 ± 0.08	0.2 ± 0.05
PLA/PEG/6CS	24 h	29.3 ± 0.72	66.3 ± 1.55	0.7 ± 0.05	3.5 ± 0.08	0.2 ± 0.04
	7 days	29.5 ± 0.65	66.1 ± 2.04	0.7 ± 0.08	3.5 ± 0.05	0.2 ± 0.05
PLA/PEG/0.5R	24 h	29.8 ± 0.27	65.7 ± 1.98	0.6 ± 0.10	3.7 ± 0.10	0.2 ± 0.05
	7 days	29.9 ± 1.63	65.4 ± 1.47	0.8 ± 0.13	3.7 ± 0.05	0.2 ± 0.05
PLA/PEG/3CS/0.5R	24 h	29.5 ± 0.45	66.1 ± 1.39	0.7 ± 0.08	3.5 ± 0.1	0.2 ± 0.04
	7 days	29.2 ± 0.69	66.2 ± 1.72	0.8 ± 0.05	3.6 ± 0.08	0.2 ± 0.05
PLA/PEG/6CS/0.5R	24 h	29.6 ± 0.33	66.0 ± 1.89	0.8 ± 0.1	3.4 ± 0.08	0.2 ± 0.05
	7 days	29.7 ± 0.47	65.9 ± 1.37	0.7 ± 0.08	3.5 ± 0.1	0.2 ± 0.05

*PMN—polymorphonuclear neutrophils, Ly—lymphocytes, E—eosinophils, M—monocytes, and B—basophils.

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
