# Peer review of "Biocompatible Materials Based on Plasticized Poly(lactic acid), Chitosan and Rosemary Ethanolic Extract I. Effect of Chitosan on the Properties of Plasticized Poly(lactic acid) Materials"

_polymers, 2019, doi:10.3390/polym11060941_

Reviewer 1 Report

I have peer reviewed the manuscript titled “Biocompatible Materials based on PEG-Plasticized 2 PLA, Chitosan and Rosemary Ethanolic Extract I. 3 Effect of chitosan on the properties of PEG-4 plasticized PLA materials” submitted to Polymers. I found this manuscript very interesting and fit well with in the scope of this journal. The manuscript needs some major improvements; there are a few suggestions that authors may consider to improve it further:

The authors have not clearly defined the aim/purpose of the study that should be included in the abstract and introduction

The title contains too many abbreviations; that may be unnecessary; please consider modifying to reduce abbreviations in title.

The abstract is unstructured; however; I could not find the statement of aim of the study: what was the aim of the study? Should be included in the abstract.

Introduction: is very detailed; and should be briefed to some extent where possible. Additionally there is lack of coherence of text in certain places. Line 51: please check for clarity and merge in to the given information.

Line 53-55: needs reference citation for this information

Line 75-76: “CS was tested in many biomedical and pharmaceutical applications, as sutures, 75 dental and bone implants and as artificial skin [28, 30, 31]” authors should cite a separate reference for each application of CS as all these references does not cover all applications; for example following references are the best suited for the regenerative and dental applications and should be included

1.      Electrospinning of chitosan-based solutions for tissue engineering and regenerative medicine. International journal of molecular sciences. 2018;19(2):407.

2.      Chitosan biomaterials for current and potential dental applications. Materials. 2017 May 31;10(6):602.

Line 84-93: are very detailed, and no reference cited; these should be briefed and reference should be cited

Line 111: reference should be 50-52.

The methodology section is well in order and detailed.

 Figure 1:  Captain is very brief; and detailed captain with labelling each SEM and description should be provided. What each image is showing? Scale bars should be bold.

Figure 3: Please present images a, b and c in a sequence; error bars should be bold.  Also, data presented in a and b can be merged as one data set

Please include limitations if any.

Author Response

Dear Reviewer,

Thanks for your valuable suggestions which help  us to improve the quality of the manuscript

Sincerely

The authors

Reviewer 2 Report

The manuscript polymers-499102 opens interesting and inspiring perspectives and deserves, in my opinion, publication in Polymers journal. However, I think that the manuscript requires some clarifications and probably reorganization to make it easier to read. Here are some suggestions:

1 Introduction is too large and, in my opinion, requires a reduction and a clearer formulation of the arguments for using poly (lactic acid) (PLA), chitosan, and rosemary extract.

1.1 Lines 34-50: material must be combined to justify the use of PLA

1.2 Lines 51-52: material is not significant for this work.

1.3 Lines 53-108: The argument for using chitosan is too broad, a lot of material that is not significant for this work.

1.4 Lines 109-150: general data are given that the reader can learn from textbooks. The article is enough to indicate the peculiarity of the composition of rosemary extract. A more visual form of representation for this is the structural formulas of key compounds.

2. Figure 1: Too much material is presented, part must be transferred to supplementary materials.

3. The structure of the composites is poorly studied, there is no data on the occurrence of reactions between components.

4. For antioxidant and antibacterial activity (section 3.7) there are no control experiments that do not allow to evaluate the level of activity of the materials presented in the current work.

5. Table 5: It is necessary to reformat the design of the table. Many mistakes and repetitions.

6. Section 3.9.2 material has a very low information value. In my opinion, it must be reduced and experimental material submitted to supplementary materials.

Author Response

(The authors gave the same response as above.)

Reviewer 3 Report

In this study, the PLA based materials were developed using natural polymer of chitosan and rosemary extract. The composites were fabricated from the different ratio of the selected materials and were subjected to a comprehensive and detailed characterization. The study is interesting and add useful information to the body of the literature and therefore worth publication. Nevertheless, I have a few suggestion/critique to make the MS clearer.

My main critique for this paper is the lack of clear explanation for the role of the additives, and their possible interactions. What is the benefit of these additives? What was the hypothesis of this manuscript? What was exactly the research question of this work? Merely mixing a range of materials cannot be very helpful and can not probably help the advancement in the field. The authors need to provide one, two paragraphs in the introduction and 1-2 sentence in the abstract to make it clear.  The authors also need to better explain the difference of the proposed composites to the materials has previously developed from chitosan. Examples are

https://www.sciencedirect.com/science/article/pii/S0141813015004730

https://doi.org/10.1016/j.ijbiomac.2015.07.012

https://www.sciencedirect.com/science/article/pii/S0141813016303543

https://doi.org/10.1016/j.ijbiomac.2016.04.046

In the abstract the author mentioned that each of the component has its own specific properties, this is vague and unclear, what are those specific properties and for what specific applications? In addition what the authors mean by high performance materials? Mechanical properties? Electrical conductivity? For what applications? The authors need to carefully revise and rewrite the abstract.

Again in line 27-28 of the abstract, it is mentioned that “The subcutaneous implantation of biocomposites induced hematological, 28 biochemical and immunological modifications comparable with control group” . this sentence is not clear and at this current form not useful.

Very importantly the authors need to revise the introduction and the whole text for brevity, the introduction in the current form composed of big block of text which make it a very hard read. This need to be reduced at least by 30%.

The design of the experiment that used (table 1) requires a better explanation. How authors decided exactly on the reported values.

Author Response

Dear Reviewer,

Thanks for your valuable suggestions which help  us to improve the quality of the manuscript

Sincerely

The authors

Round  2

Reviewer 1 Report

“Many thanks for the revision and incorporating all suggested changes to the manuscript”

Reviewer 2 Report

The second version of the manuscript polymers-499102 can be published in present form.